

# Comparison of efficacy and costs between robotic-assisted and conventional thoracoscopic approaches for partial pulmonary resection: a systematic review and meta-analysis of propensity score-matched studies

Xinyang Huang[1,*], Haoxuan Li[2,*], Zihao Deng[1], Xinyuan Tian[1], Kunjiang Zhong[1] and Xugang Zhang[1]

[1] Beijing Shijitan Hospital, Capital Medical University, Beijing, China
[2] School of Basic Medicine, Capital Medical University, Beijing, China
[*] These authors contributed equally to this work.

Corresponding author
Xugang Zhang,
zhangxugang2010@126.com

## ABSTRACT

**Background**. This meta-analysis aimed to evaluate differences in perioperative outcomes and costs between robotic-assisted partial pulmonary resection (RAPPR) and video-assisted thoracoscopic partial pulmonary resection (VATPPR).

**Methods**. We systematically searched MEDLINE, PubMed, Google Scholar, and Cochrane databases for relevant studies published between March 2015 and March 2025. Propensity score-matched non-randomized controlled studies comparing RAPPR with VATPPR were included.

**Results**. Eight propensity score-matched studies involving 3,104 patients were included: 1,528 patients underwent RAPPR and 1,576 underwent VATPPR. RAPPR was associated with longer operative time and higher medical costs, but patients had more lymph nodes dissected, shorter drainage tube duration, and shorter hospital length of stay. No significant differences were observed between the two groups in conversion to thoracotomy rates or complication rates, including persistent air leak, pneumonia, and chylothorax.

**Conclusions**. RAPPR demonstrates comparable surgical efficacy to VATPPR with advantages including more thorough lymph node dissection, earlier drainage tube removal, and earlier patient discharge. However, RAPPR requires longer operative time and higher costs. The choice between surgical approaches should consider these clinical factors comprehensively.

## INTRODUCTION

Partial pulmonary resection (PPR) encompasses various procedures involving localized lung tissue removal. These include lobectomy, segmentectomy, and wedge resection (*Zhang et al., 2024b*). PPR serves as an effective treatment modality for non-small cell lung cancer and certain benign pulmonary diseases. The minimally invasive evolution of PPR has substantially enhanced perioperative patient experience (*Khan et al., 2024*).

Video-assisted thoracoscopic surgery was introduced into clinical practice in the 1990s. It became the conventional minimally invasive approach for partial pulmonary resection. This technique offers advantages of reduced trauma, fewer complications, and faster recovery (*Napolitano et al., 2022*). However, the landscape began changing in 2002 when the first robotic lobectomy for primary lung cancer was reported (*Melfi et al., 2002*). The widespread adoption of the Da Vinci surgical system has since resulted in rapidly expanding applications of robotics in thoracic minimally invasive surgery.

The robotic surgical system offers several technological advantages. These include three-dimensional high-definition visualization, tremor filtration, seven-degree-of-freedom instrument articulation, and 360° robotic arm rotation (*Ochi et al., 2023*; *Ureña et al., 2023*; *Rabinovics & Aidan, 2015*). These features enable more precise operations in partial pulmonary resection. Theoretically, they provide further optimization of surgical safety and efficacy. Consequently, robotic approaches are gradually gaining recognition among surgeons.

Although studies have compared the efficacy of robot-assisted partial pulmonary resection (RAPPR) with video-assisted thoracoscopic partial pulmonary resection (VATPPR), limitations exist. Most literature consists of single-center small case series. These studies lack propensity score-matched (PSM) analysis, making it difficult to draw broadly applicable conclusions. To address this gap, we conducted this updated meta-analysis. Our objective was to compare differences in perioperative outcomes and costs between RAPPR and VATPPR. This study incorporated the most comprehensive and current literature available to date. We implemented strict screening for PSM studies, thereby enhancing the reliability and scientific rigor of our findings.

## MATERIALS AND METHODS

### Literature search criteria

This study adhered to the Preferred Reporting Items for Systematic Reviews and Meta-Analyses (PRISMA) guidelines (*Page et al., 2021*). The protocol was prospectively registered in the PROSPERO database (CRD420251009131). All propensity score-matched studies comparing RAPPR and VATPPR met inclusion criteria. These studies examined efficacy and costs in partial pulmonary resection procedures. Included procedures encompassed lobectomy, segmentectomy, and wedge resection.

Between March 29 and March 30, 2025, two investigators conducted comprehensive searches. Xinyang Huang and Haoxuan Li independently searched multiple databases. These included MEDLINE, PubMed, Google Scholar, Cochrane Library, and clinical trial registries. The objective was to identify English-language PSM studies involving RAPPR

and VATPPR. Search terms included "robotic-assisted", "video-assisted thoracoscopic surgery", "lobectomy", "segmentectomy", and "propensity score matching". These terms were combined using Boolean operators AND/OR. Beyond manual literature searches, we performed secondary reference screening. We also conducted further analysis of additional eligible studies.

## Inclusion and exclusion criteria

This study employed the PICOS framework to establish inclusion criteria. PICOS represents Population, Intervention, Comparison, Outcomes, and Study design. Population (P) included patients undergoing PPR. Intervention (I) was RAPPR. Comparison (C) was VATPPR. Outcomes (O) encompassed multiple measures. Intraoperative outcome measures included operative time, number of lymph nodes dissected, and conversion to thoracotomy rate. Postoperative outcome measures included drainage tube indwelling time, length of hospital stay, overall complication incidence, persistent air leak incidence, pneumonia incidence, and chylothorax incidence. Cost measures included surgical expenses. Study design (S) required non-randomized controlled studies implementing PSM. Eligible studies needed to report at least four postoperative outcomes.

Exclusion criteria included several categories. These were non-English publications, conference abstracts or letters, studies analyzing outcomes following surgery as adjuvant or combination therapy, studies with patients receiving other surgical interventions (such as radiofrequency ablation), non-PSM case-control studies, and animal experimental studies.

## Study selection and data collection

Two investigators performed screening according to inclusion and exclusion criteria. Xinyang Huang and Haoxuan Li worked independently. Initially, they screened potentially eligible studies by analyzing article titles and abstracts. Subsequently, team members independently reviewed the full text of each qualifying article. When disagreements arose during screening, investigators engaged in discussion. They analyzed disputed literature strictly according to inclusion and exclusion criteria item by item. When necessary, they sought input from the referee (Xugang Zhang). Alternatively, they convened team meetings for collective discussion to resolve disputes and reach consensus.

## Data extraction and management

Two team members extracted data from eligible studies. They entered information into Excel spreadsheets. Key extracted data included multiple categories. These encompassed first author, publication year, country of origin, patient demographic characteristics, baseline pulmonary function parameters, perioperative outcome measures, and surgical costs.

## Statistical analysis

This study employed Review Manager V5.3.1 software for statistical analysis. Results were expressed with 95% confidence intervals (CI). Odds ratios (OR) were used for dichotomous variables. Weighted mean differences (WMD) were used for continuous variables. For data lacking means (M) and standard deviations (SD), we employed the method by *Luo et al. (2018)*. This method converted these values to means and SD.

Dichotomous variables were analyzed using the Mantel-Haenszel method. Continuous variables were analyzed using the inverse variance method. All analyses employed random-effects models. This approach considered potential substantial heterogeneity between different studies. Heterogeneity was assessed using the $I^2$ statistic. Values of 0%–40% represented low heterogeneity. Values of 30%–60% indicated moderate heterogeneity. Values of 50%–90% showed high heterogeneity. Values of 75%–100% represented considerable heterogeneity that cannot be ignored. Statistical significance was set at $P < 0.05$.

Since all included studies were cohort studies, study quality assessment employed the ROBINS-I tool. Additionally, we conducted sensitivity analyses for outcome measures with significant heterogeneity. These analyses examined the robustness of conclusions. Given that the final number of included studies did not exceed 10, statistical power was relatively low. Therefore, we did not proceed with publication bias analysis (*Sterne, Gavaghan & Egger, 2000*; *Lau et al., 2006*).

# RESULTS

## Baseline characteristics

Based on our literature search strategy and inclusion criteria, eight studies met eligibility criteria and were included in the meta-analysis (*Chen et al., 2022*; *Yang et al., 2018*; *Wu & Ma, 2023*; *Lan et al., 2024*; *Yang et al., 2017*; *Zhang et al., 2024a*; *Gómez-Hernández et al., 2024*; *Zhou et al., 2024*). Table 1 summarizes the main characteristics, perioperative outcomes, and surgical costs of these studies. The studies encompassed 3,104 patients total. Among these, 1,528 patients underwent RAPPR and 1,576 patients received conventional VATPPR. Figure 1 presents the study selection process based on the PRISMA flow diagram. Table 2 lists comparative baseline characteristic data from these studies.

Our analysis revealed no significant difference in the proportion of male patients between RAPPR and VATPPR groups ($P = 0.46$). Additionally, age ($P = 0.49$), BMI ($P = 0.60$), smoking rate ($P = 0.50$), and FEV1% ($P = 0.26$) showed no statistical differences. These findings indicate good comparability between the two groups regarding baseline characteristics.

## Quality assessment

This study employed the ROBINS-I tool to assess the quality of included cohort studies. Assessment results showed that *Chen et al. (2022)* was rated as high risk. *Zhou et al. (2024)* and *Wu & Ma (2023)* were rated as moderate risk. Other studies were rated as low risk (detailed quality assessment results are shown in Fig. 2).

Although *Chen et al. (2022)* carries higher bias risk, it comprehensively reported perioperative outcomes and cost indicators of interest to our study (Table 1). Its relatively large sample size makes it still valuable for inclusion. We will interpret results related to this study with caution. We will prioritize its performance in sensitivity analyses and its impact on overall conclusion robustness.

**Table 1  Characteristics studied and perioperative outcomes.**

| | Chen2022 | | Yang2018 | | Wu2023 | | Lan2024 | | Yang2017 | | Zhang2024 | | Gómez-Hernández2024 | | Zhou2024 | |
|---|---|---|---|---|---|---|---|---|---|---|---|---|---|---|---|---|
| Country | China | | China | | China | | China | | USA | | China | | Spain | | China | |
| Reference No. | *Chen et al. (2022)* | | *Yang et al. (2018)* | | *Zhiqiang & Shaohua (2023)* | | *Lan et al. (2024)* | | *Yang et al. (2017)* | | *Zhang et al. (2024a)* | | *Gómez-Hernández et al. (2024)* | | *Zhou et al. (2024)* | |
| Surgical method | RAPPR | VATPPR | RAPPR | VATPPR | RAPPR | VATPPR | RAPPR | VATPPR | RAPPR | VATPPR | RAPPR | VATPPR | RAPPR | VATPPR | RAPPR | VATPPR |
| Patient | 107 | 144 | 69 | 69 | 71 | 71 | 42 | 84 | 172 | 141 | 148 | 148 | 73 | 73 | 846 | 846 |
| Male | 53 (49.5%) | 72 (50.0%) | 24 (34.8%) | 24 (34.8%) | 30 (42.3%) | 29 (40.8%) | 23 (54.8%) | 40 (47.6%) | 74 (43.0%) | 53 (37.6%) | 47 (31.8%) | 52 (35.1%) | 42 (57.5%) | 43 (58.9%) | 419 (49.5%) | 406 (48.0%) |
| Age (year) | 69.80 ± 4.10 | 69.50 ± 3.70 | 59.51 ± 8.87 | 59.54 ± 10.04 | 59.95 ± 11.35 | 60.29 ± 9.08 | 58.10 ± 9.40 | 58.10 ± 11.50 | 68.00 ± 10.20 | 67.50 ± 10.00 | 55.00 ± 8.98 | 56.74 ± 12.54 | 66.82 ± 7.94 | 67.35 ± 10.59 | 57.50 ± 9.60 | 58.00 ± 9.00 |
| BMI (kg/m$^2$) | 23.80 ± 3.50 | 23.70 ± 2.80 | NA | NA | 23.63 ± 2.64 | 23.93 ± 2.98 | 22.90 ± 2.50 | 23.40 ± 3.20 | NA | NA | 23.24 ± 2.40 | 22.82 ± 2.50 | 26.56 ± 4.36 | 26.44 ± 4.25 | NA | NA |
| Smoking history | 44 (41.1%) | 58 (40.3%) | NA | NA | 54 (76.1%) | 55 (77.5%) | 12 (28.6%) | 19 (22.6%) | 139 (80.8%) | 115 (81.6%) | 30 (20.3%) | 33 (22.3%) | NA | NA | 405 (47.9%) | 390 (46.1%) |
| FEV1% | 77.30 ± 9.00 | 76.10 ± 10.30 | 77.32 ± 5.02 | 78.81 ± 4.89 | NA | NA | NA | NA | 91.60 ± 17.40 | 90.30 ± 17.90 | 96.86 ± 15.14 | 90.10 ± 14.71 | 90.81 ± 18.09 | 95.11 ± 29.50 | NA | NA |
| Operative time (min) | 120.80 ± 35.00 | 165.10 ± 54.10 | 148.95 ± 36.83 | 137.50 ± 38.08 | 143.23 ± 31.78 | 134.00 ± 40.11 | 170.64 ± 49.41 | 137.84 ± 52.33 | NA | NA | 94.57 ± 7.59 | 85.34 ± 9.11 | 123.53 ± 37.81 | 110.58 ± 52.94 | 138.80 ± 61.80 | 132.80 ± 43.20 |
| Number of lymph node dissected | 12.20 ± 5.70 | 8.10 ± 5.00 | 12.67 ± 5.59 | 12.01 ± 6.21 | NA | NA | 13.54 ± 9.65 | 13.69 ± 6.36 | NA | NA | 10.65 ± 5.24 | 8.35 ± 5.24 | NA | NA | NA | NA |
| Conversion to thoracotomy | 0 (0.0%) | 28 (19.4%) | NA | NA | 0 (0.0%) | 8 (11.3%) | NA | NA | 16 (9.3%) | 8 (5.7%) | NA | NA | 2 (2.7%) | 0 (0.0%) | 10 (1.2%) | 43 (5.1%) |
| Indwelling time of drainage tube (day) | 6.20 ± 3.70 | 7.70 ± 5.30 | 2.07 ± 1.09 | 1.67 ± 1.36 | 3.35 ± 0.76 | 4.35 ± 2.27 | 3.49 ± 2.07 | 3.57 ± 2.67 | NA | NA | 3.00 ± 0.38 | 3.00 ± 0.38 | NA | NA | 3.60 ± 2.70 | 4.10 ± 2.40 |
| Length of hospital stay (day) | 8.60 ± 3.90 | 10.80 ± 5.40 | 3.94 ± 1.00 | 3.81 ± 1.52 | 4.00 ± 1.51 | 4.71 ± 1.51 | 4.37 ± 3.68 | 4.63 ± 2.87 | 4.97 ± 5.77 | 5.96 ± 9.17 | 7.35 ± 2.25 | 7.18 ± 1.87 | 2.65 ± 0.76 | 3.18 ± 1.13 | 11.40 ± 4.90 | 10.50 ± 3.70 |
| Overall complications | 26 (24.3%) | 36 (25.0%) | 2 (2.9%) | 4 (5.8%) | NA | NA | 7 (16.7%) | 12 (14.3%) | 51 (29.7%) | 35 (24.8%) | 19 (12.8%) | 25 (16.9%) | NA | NA | NA | NA |
| Persistent air leak | 11 (10.3%) | 17 (11.8%) | NA | NA | 1 (1.4%) | 14 (19.7%) | 3 (7.1%) | 6 (7.1%) | NA | NA | 5 (3.4%) | 3 (2.0%) | 3 (4.1%) | 4 (5.5%) | 38 (4.5%) | 54 (6.4%) |
| Pneumonia | 6 (5.6%) | 20 (13.9%) | NA | NA | NA | NA | 1 (2.4%) | 0 (0.0%) | NA | NA | 8 (5.4%) | 4 (2.7%) | NA | NA | 1 (0.1%) | 1 (0.1%) |
| Chylothorax | 0 (0.0%) | 3 (2.1%) | NA | NA | 4(5.6%) | 0 (0.0%) | 0(0.0%) | 0 (0.0%) | NA | NA | 1 (0.7%) | 1 (0.7%) | NA | NA | 12 (1.4%) | 12 (1.4%) |
| Cost ($10000) | 1.80 ± 0.31 | 1.48 ± 0.35 | NA | NA | NA | NA | 1.58 ± 0.15 | 1.23 ± 0.26 | NA | NA | 1.02 ± 0.02 | 0.69 ± 0.04 | NA | NA | 1.01 ± 0.20 | 0.67 ± 0.27 |

**Notes.**

M ± SD, Representation of continuous variables; n (%), representation of discrete variables; M, mean; SD, standard deviation; RAPPR, robot assisted partial pulmonary resection; VATPPR, video assisted thoracoscopic partial pulmonary resection; BMI, body mass index; FEV1, forced expiratory volume in 1 second; NA, not available.
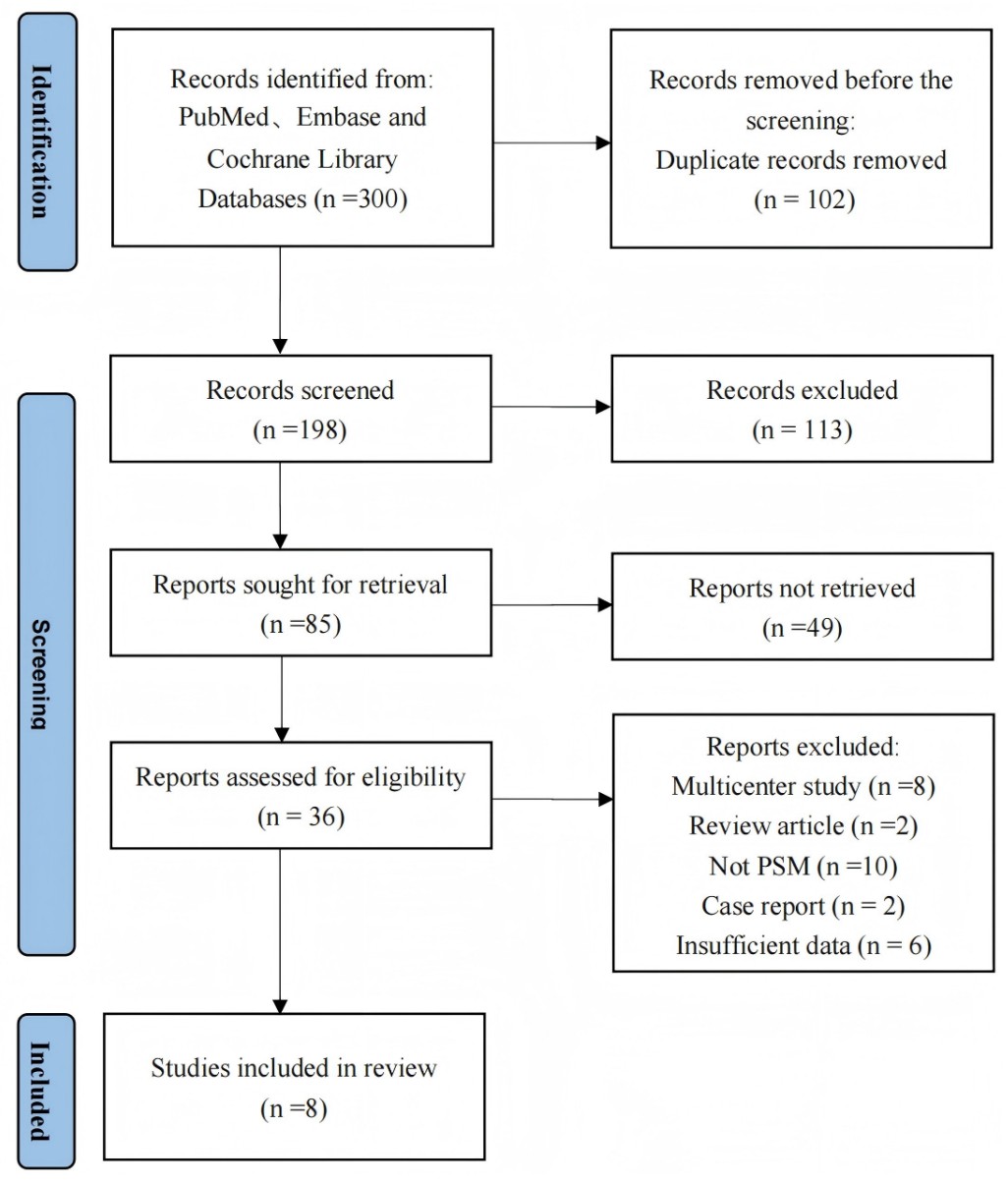

**Figure 1   The PRISMA flowchart.**

## Meta-analysis and sensitivity analysis of intraoperative outcome measures

Through meta-analysis, we found no significant difference in operative time between RAPPR and VATPPR (WMD 4.47, 95% CI [−6.87–15.81], $P = 0.44$, $I^2 = 94\%$) (Fig. SA). However, significant heterogeneity was noted for this indicator. We performed "leave-one-out" sensitivity analysis. After excluding *Chen et al. (2022)*, heterogeneity decreased to low levels. RAPPR demonstrated significantly longer operative time (WMD 9.69, 95% CI [5.81–13.57], $P < 0.00001$, $I^2 = 38\%$) (Fig. 3A).
**Table 2  The demographics of the studies.**

| Variable | Number of studies with available data | WMD/OR | 95% CI | P-value |
|---|---|---|---|---|
| Male (n) | 8 | 1.06 | (0.91–1.22) | 0.46 |
| Age (years) | 8 | −0.20 | (−0.79–0.38) | 0.49 |
| BMI (kg/m$^2$) | 5 | 0.10 | (−0.27–0.47) | 0.60 |
| Smoking history (n) | 6 | 1.06 | (0.90–1.23) | 0.50 |
| FEV1 % | 4 | 2.07 | (−1.51–5.65) | 0.26 |

Notes.
WMD, weighted mean difference; OR, odds ratio; Cl, confidence interval; BMI, body mass index; FEV1, forced expiratory volume in 1 s.

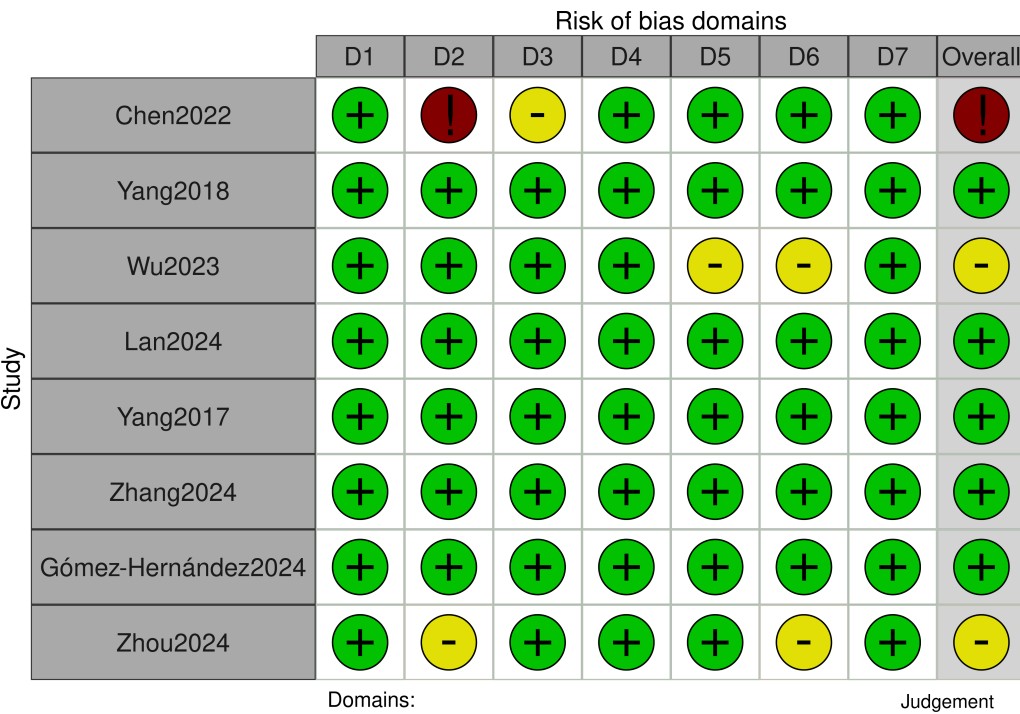

**Figure 2  ROBINS-I tool: quality evaluation ChartROBINS-I.** *Chen et al., 2022*; *Yang et al., 2018*; *Wu & Ma, 2023*; *Lan et al., 2024*; *Yang et al., 2017*; *Zhang et al., 2024a*; *Gómez-Hernández et al., 2024*; *Zhou et al., 2024*.

Furthermore, RAPPR achieved significantly more lymph node dissection (WMD 2.05, 95% CI [0.35–3.74], $P = 0.02$, $I^2 = 74\%$) (Fig. 3B). Finally, RAPPR showed no significant difference in conversion to thoracotomy rate (OR 0.33, 95% CI [0.06–1.70], $P = 0.19$, $I^2 = 83\%$) (Fig. SB). The significant heterogeneity indicated that sensitivity
**A**

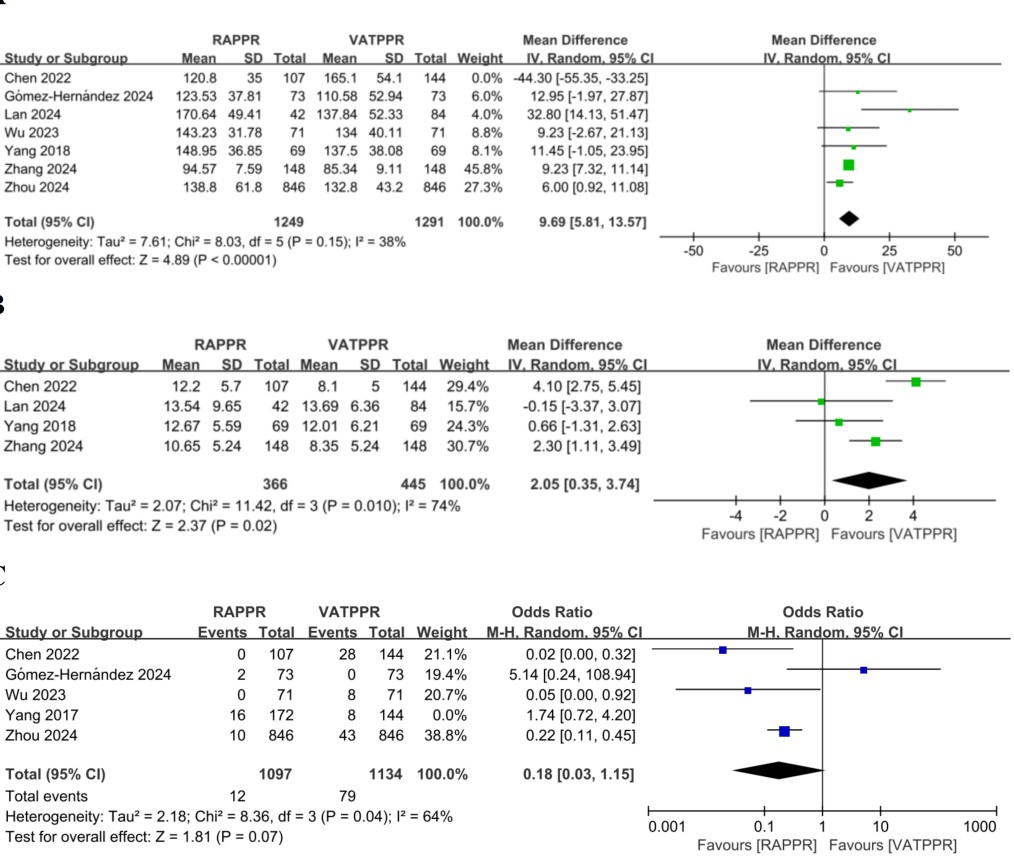

**B**

**C**

> **Figure 3** (A–C) **Forest plots for intraoperative outcome measures.** Studies: *Chen et al. (2022)*; *Gómez-Hernández et al. (2024)*; *Lan et al. (2024)*; *Wu & Ma (2023)*; *Yang et al. (2018)*; *Zhang et al. (2024a)*; *Zhou et al. (2024)*.
>
>

analysis was necessary. "Leave-one-out" results showed that after excluding *Yang et al. (2017)*, heterogeneity improved. Both surgical approaches still showed no significant difference in conversion to thoracotomy rates (OR 0.18, 95% CI [0.03–1.15], $P = 0.07$, $I^2 = 64\%$) (Fig. 3C). However, the source of heterogeneity remained unclear.

## Meta-analysis and sensitivity analysis of postoperative outcome measures

Pooled results indicated no significant difference in drainage tube indwelling time for RAPPR compared to conventional thoracoscopy (WMD −0.33, 95% CI [−0.72–0.06], $P = 0.10$, $I^2 = 86\%$) (Fig. SC). Heterogeneity was significant. After sensitivity analysis excluding *Yang et al. (2018)* and *Zhang et al. (2024a)*, heterogeneity improved. RAPPR demonstrated significantly shorter drainage tube indwelling time (WMD −0.68, 95% CI [−1.13 to −0.24], $P = 0.002$, $I^2 = 54\%$) (Fig. 4A).

Additionally, RAPPR showed no significant difference in length of hospital stay (WMD −0.30, 95% CI [−0.84–0.24], $P = 0.28$, $I^2 = 86\%$) (Fig. SD). However, sensitivity analysis conducted due to significant heterogeneity showed improved heterogeneity after

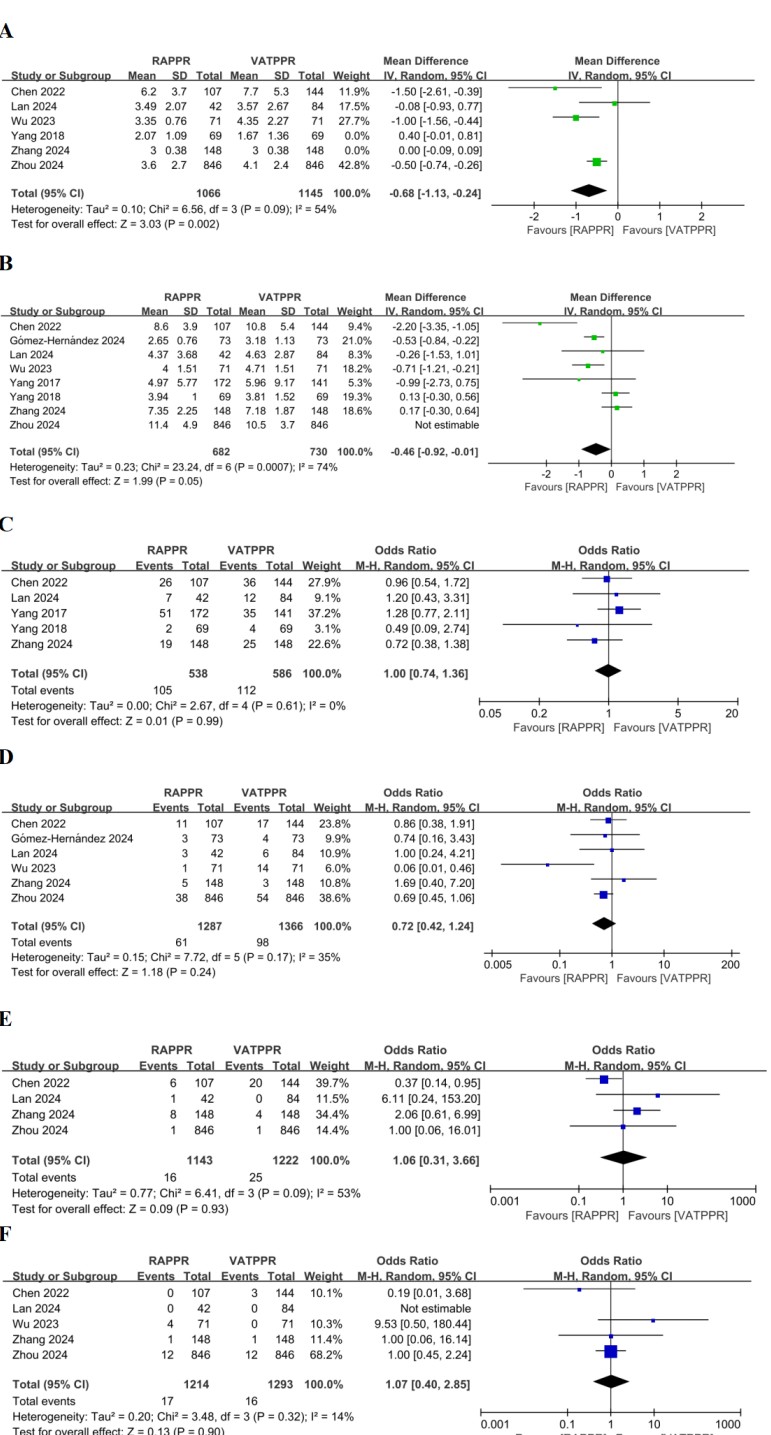

**Figure 4** (A–F) Forest plot for postoperative outcome measures. Studies: *Chen et al. (2022)*; *Lan et al. (2024)*; *Wu & Ma (2023)*; *Yang et al. (2018)*; *Yang et al. (2018)*; *Zhang et al. (2024a)*; *Zhou et al. (2024)*; *Gómez-Hernández et al. (2024)*; *Yang et al. (2018)*; *Yang et al. (2017)*.

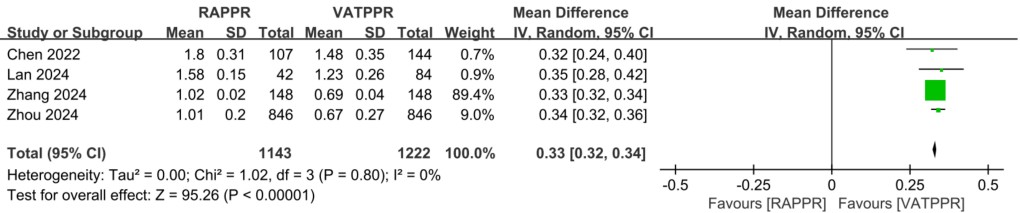

**Figure 5** **Forest plot for medical cost.** Studies: *Chen et al. (2022)*; *Lan et al. (2024)*; *Zhang et al. (2024a)*; *Zhou et al. (2024)*.

excluding *Zhou et al. (2024)*. RAPPR demonstrated significantly shorter hospital stay (WMD $-0.46$, 95% CI [$-0.92$ to $-0.01$], $0.04 < P < 0.05$, $I^2 = 74\%$) (Fig. 4B).

Regarding complications, RAPPR showed no significant differences in overall complication incidence (OR 1.00, 95% CI [0.74–1.36], $P = 0.99$, $I^2 = 0\%$) (Fig. 4C), persistent air leak incidence (OR 0.72, 95% CI [0.42–1.23], $P = 0.23$, $I^2 = 35\%$) (Fig. 4D), pneumonia incidence (OR 1.06, 95% CI [0.31–3.66], $P = 0.93$, $I^2 = 53\%$) (Fig. 4E), and chylothorax incidence (OR 1.07, 95% CI [0.40–2.85], $P = 0.90$, $I^2 = 14\%$) (Fig. 4F). Heterogeneity performance was satisfactory for these measures.

## Meta-analysis of cost measures

Pooled results from four studies showed that RAPPR incurred significantly higher medical costs compared to conventional thoracoscopy (WMD 0.33, 95% CI [0.32–0.34], $P < 0.00001$, $I^2 = 0\%$) (Fig. 5).

## DISCUSSION

In our initial analysis, RAPPR and VATPPR showed no difference in operative time. However, after sensitivity analysis excluding *Chen et al. (2022)*, heterogeneity significantly improved and revealed that RAPPR demonstrated significantly prolonged operative time. Further analysis indicated that in *Chen et al. (2022)*, 5.6% of patients in the thoracoscopic group had chronic respiratory comorbidities (including COPD, asthma, and silicosis), while only 2.8% in the robotic group had these conditions (*Chen et al., 2022*). This represents a significant selection bias and constitutes a major reason why we classified this study as high-risk bias. Taking COPD as an example, patient lung tissue often exhibits emphysema or diffuse parenchymal destruction. This leads to increased lung tissue fragility, potentially causing tissue tears or air leaks during cutting or suturing. These conditions require more meticulous operations and additional time for repair. Intraoperative single-lung ventilation management also becomes more complex.

Regarding operative time, two studies support our conclusions (*Hu et al., 2020*; *Hu & Wang, 2019*), suggesting that robotic lung resection requires more time. One important factor is that RAPPR requires additional time beyond the main operative period for robotic system setup and calibration (averaging approximately $10.2 \pm 4.0$ min). This phase includes equipment startup and positioning, instrument connection, and other preparatory steps,

directly increasing non-operative time (*Wilson-Smith et al., 2023*). Additionally, operative time closely correlates with the learning curve of robotic surgery. *Wilson-Smith et al. (2023)* found that technical proficiency for robotic lobectomy averaged $25.3 \pm 12.6$ cases, with mean operative time of $190.5 \pm 53.8$ minutes. A Japanese study showed that as surgeons gained experience performing RAPPR, main operative time decreased from 171 min to 149 minutes (*Haruki et al., 2025*). Another Italian study suggested that experience accumulation could shorten the learning curve for single-port RAPPR by 52 minutes (*Mercadante et al., 2022*). These data demonstrate that thoracic surgeons require specific time and case numbers to achieve efficient robotic surgery performance.

Finally, PPR complexity represents a key factor influencing time differences and robotic advantage manifestation, requiring individualized assessment. For patients undergoing simple PPR procedures, prolonged robotic operative time may primarily increase anesthetic risks. However, for complex segmentectomies (such as intrinsic upper lobe apical posterior segment S1+2 or basal segment combination resections) or bronchial/vascular sleeve reconstructions and other highly complex PPR procedures, robotic assistance enables precise dissection in deep, narrow spaces and accurate division and anastomosis of small vessels and bronchioles (*Wada et al., 2025*; *Watkins, Quadri & Servais, 2021*). This significantly reduces lung tissue injury risk, controls bleeding, and decreases perioperative complications (*Wu et al., 2021*; *Zhou et al., 2022*). These potential advantages may outweigh the limitations of prolonged operative time.

This meta-analysis demonstrated that RAPPR achieved significantly more lymph node dissection compared to conventional thoracoscopy. *Chen et al. (2022)* (12.20 *vs.* 8.10, $P < 0.0001$) and *Zhang et al. (2024a)* (10.65 *vs.* 8.35, $P = 0.0002$) also supported this finding. Another study focusing on long-term lung cancer survival found increased lymph node retrieval in the robotic group (11.75 *vs.* 9.77, $P < 0.001$). The number of nodes dissected showed significant correlation with overall survival (OR 1.94, 95% CI [1.07–3.51], $P = 0.029$) (*Zhang et al., 2025*). This may be attributed to the Da Vinci robotic surgical system's three-dimensional visualization, which better visualizes intrathoracic anatomical structures. The camera can move between different ports, precisely locating lymph node positions. The more flexible robotic arms can achieve greater lymph node dissection with high degrees of freedom within the confined thoracic cavity. This particularly applies to areas difficult to reach with conventional thoracoscopy, such as mediastinal lymph nodes (*Ureña et al., 2023*; *Casiraghi et al., 2024*).

More thorough lymph node dissection holds significant importance for surgical treatment of pulmonary malignancies. It can improve lung cancer staging accuracy (*Handa et al., 2023*) and helps prevent lung cancer recurrence and lymph node metastasis, potentially enhancing patient overall survival. However, it must be emphasized that not all PPR procedures require extensive lymph node dissection. For lung nodule resections definitively identified as benign lesions, or wedge resections for pure ground-glass opacities (GGO) with intraoperative frozen pathology confirming adenocarcinoma *in situ* (AIS) or minimally invasive adenocarcinoma (MIA), RAPPR's lymph node dissection advantages may not translate to significant clinical benefits. In these cases, conventional VATPPR can serve as an excellent treatment option.

Both surgical approaches showed no significant difference in conversion to thoracotomy rates, consistent with previous conclusions by *Mao et al. (2021)*. This suggests robotic technology does not significantly reduce intraoperative emergency conversion to thoracotomy requirements. However, *Tasoudis et al. (2023)* concluded that the robotic group had higher conversion rates. This may be because their study included literature where robotic approaches were more frequently used for early peripheral lung cancers. Complex central lesions using robotic surgery present higher difficulty and may increase conversion to thoracotomy risk. For patients, the absence of increased conversion rates indicates that both robotic and thoracoscopic groups maintain high safety levels. Both approaches positively impact postoperative patient recovery.

Our study found that RAPPR drainage tube indwelling time was significantly shortened after excluding two studies with higher proportions of upper lobe tumors. This finding aligns with *Ma et al. (2021)* conclusions. Detailed analysis of these two studies reveals that in *Yang et al. (2018)*, the proportion of patients with upper lobe tumor location was 69.57% in the robotic group *versus* 52.17% in the thoracoscopic group. Similarly, in *Zhang et al. (2024a)*, these proportions were 66.22% *versus* 57.43%, respectively. Compared to other lobectomies, upper lobectomy results in more significant postoperative pulmonary function parameter declines (such as FEV1, FVC). This may delay pulmonary function recovery and slow pleural effusion absorption, thereby prolonging drainage tube indwelling time (*Fukui et al., 2020*).

However, in *Emmert et al. (2017)*, VATPPR demonstrated shorter drainage time than RAPPR. This contradiction may arise from robotic arm rigidity potentially increasing intrathoracic tissue traction injury risk. Compared to conventional thoracoscopic instruments, robotic arms lack tactile feedback. Surgeons may experience force control deviations during adhesion separation or lung lobe traction, causing minor bronchial tears or pleural injury. This leads to increased postoperative fluid extravasation, requiring prolonged drainage tube indwelling time. However, in our study, the robotic group showed shorter drainage time. This advantage means earlier tube removal, reduced hospitalization time, and decreased infection risk for patients, particularly benefiting populations with poor baseline pulmonary function.

In our initial analysis, RAPPR and VATPPR showed no difference in length of hospital stay. However, after sensitivity analysis excluding *Zhou et al. (2024)*, heterogeneity significantly improved and demonstrated shorter hospital stay for RAPPR patients. In the Zhou 2024 study, 9.7% of robotic group patients received preoperative neoadjuvant therapy compared to 4.5% in the thoracoscopic group. This prolonged hospitalization time (*Zhou et al., 2024*) and created substantial heterogeneity in the pooled effect. *Chen et al. (2022)* found that compared to conventional thoracoscopy, robotic surgery showed less estimated blood loss (69.80 *vs.* 136.50, $P < 0.00001$) and significantly reduced hospital stay (8.60 *vs.* 10.80, $P = 0.0002$). These significant reductions were also evident in *Qiu et al. (2020)*.

The reason may be that robotic surgery demonstrates significantly improved operative precision and eliminates hand tremor. This reduces accidental injury to surrounding vessels and tissues and provides better control of small vessel bleeding. Reduced blood loss and

low trauma decrease patient postoperative transfusion requirements and infection risk (*De Vermandois et al., 2019*). This also reduces postoperative pain (*Zhang et al., 2024a*), promotes early ambulation and pulmonary function recovery, thereby shortening hospital stay. Faster patient healing and discharge can reduce postoperative care burden and costs, benefiting overall public health and socioeconomic outcomes (*Handa, Gaidhane & Choudhari, 2024*).

Regarding complications, pooled results indicated no significant differences between RAPPR and VATPPR. This included overall incidence, persistent air leak, pneumonia, and chylothorax. *Lan et al. (2024)* and *Zhou et al. (2024)* reached identical conclusions. However, *Ueno et al. (2024)* found significantly higher persistent air leak incidence in robotic surgery within their cohort (17 *vs.* 6 cases, $P = 0.02$). This may be because robotic end-effectors have limited tactile feedback. Compared to conventional thoracoscopy, accidental contact between surgical instruments and residual lung tissue may occur more frequently, leading to undetected lung injury. Additionally, robotic surgery commonly uses energy devices such as electrocautery and ultrasonic scalpels for tissue cutting and hemostasis. High temperatures from energy devices may cause thermal injury to lung tissue margins (*Shibao et al., 2021*). This weakens healing capacity at cut edges, leading to persistent postoperative air leak formation. Therefore, comparative results regarding persistent air leak and other complications still require more high-quality randomized controlled trials for further verification.

Our study demonstrated that RAPPR incurred significantly higher medical costs compared to conventional thoracoscopy. *Lan et al. (2024)* showed consistent results (1.58 *vs.* 1.23, $P < 0.00001$). This finding is not surprising, as robotic technology application in thoracic surgery frequently faces cost-related criticism. Currently, a complete Da Vinci surgical system requires initial investment exceeding 10 million yuan RMB. Annual maintenance costs range from $100,000 to $150,000 (*Patel et al., 2023*). Single surgery instrument and consumable costs are also higher. Recurring expenses for disposable items range from $400 to $1,200. Robotic surgery-specific consumable components, including scissors, graspers, needle holders, and staplers, are considered the primary cause of overall cost differences between the two surgical approaches (*Shanahan et al., 2022*).

Furthermore, longer robotic operating room time represents an important factor. Patient surgery costs, including anesthesia, are estimated at $40 per minute *Childers & Maggard-Gibbons, 2018*). Based on this, *Tupper et al. (2024)* indicated that robotic-assisted lobectomy time costs are estimated to be $824 higher. However, multiple studies suggest that robotic approaches may partially offset intraoperative high costs through postoperative advantages. These include reduced opioid analgesic use, decreased transfusion requirements, lower infection risk and antibiotic use, and faster return to work (*Bastawrous et al., 2022*; *Bijlani et al., 2016*; *Shkolyar et al., 2020*). Nevertheless, more data remain necessary for comprehensive RAPPR cost-benefit evaluation.

Similar to other retrospective studies, our study limitations include retrospective design. Although robotic and thoracoscopic groups demonstrated good comparability after PSM, selection bias persists. Additionally, *Chen et al.*'s (*2022*) high-bias-risk study was included. Although this study employed PSM, potential unmeasured confounding factors

remain concerning. We addressed this issue through sensitivity analysis, which confirmed robustness of relevant outcome measures (such as operative time) after excluding this study. Furthermore, limited included literature prevented in-depth exploration of differences between lung resection types (lobectomy, segmentectomy, wedge resection). We also could not perform subgroup analyses on demographic baseline factors, which would help analyze heterogeneity sources.

Finally, efficacy evaluation of different surgical methods should not be limited to these short-term perioperative outcomes. More long-term survival data are also necessary, but included studies did not provide such information. Therefore, we cannot comprehensively evaluate long-term efficacy differences between RAPPR and VATPPR. In conclusion, future research still requires more high-quality studies for further exploration.

## CONCLUSIONS

This study demonstrates that RAPPR achieves comparable surgical efficacy to conventional VATPPR. RAPPR offers specific advantages including more thorough lymph node dissection, shorter drainage tube indwelling time, and reduced length of hospital stay. However, RAPPR requires longer operative time and incurs higher medical costs.

To enhance the accuracy of our conclusions regarding perioperative outcomes and costs, and to further evaluate the long-term efficacy of RAPPR, future randomized controlled trials with extended follow-up periods are necessary.

## ACKNOWLEDGEMENTS

Special thanks to Beijing Shijitan Hospital, Capital Medical University for its strong support for this study. In addition, special thanks to Professor Liu Lei of the hospital science and technology department for his guidance and suggestions on the methodology of this study.

### Funding

This work was supported by Chinese National Railway Group's 2023 Medical and Health Special Scientific Research Project Plan (J2023Z613). The funders had no role in study design, data collection and analysis, decision to publish, or preparation of the manuscript.

### Grant Disclosures

The following grant information was disclosed by the authors:
Chinese National Railway Group's 2023 Medical and Health Special Scientific Research Project Plan: J2023Z613.

### Competing Interests

The authors declare there are no competing interests.

## Author Contributions

- Xinyang Huang conceived and designed the experiments, performed the experiments, analyzed the data, prepared figures and/or tables, authored or reviewed drafts of the article, and approved the final draft.
- Haoxuan Li conceived and designed the experiments, performed the experiments, analyzed the data, prepared figures and/or tables, authored or reviewed drafts of the article, and approved the final draft.
- Zihao Deng analyzed the data, prepared figures and/or tables, and approved the final draft.
- Xinyuan Tian performed the experiments, prepared figures and/or tables, and approved the final draft.
- Kunjiang Zhong conceived and designed the experiments, prepared figures and/or tables, and approved the final draft.
- Xugang Zhang analyzed the data, authored or reviewed drafts of the article, and approved the final draft.

## Data Availability

Raw data is available in Tables 1 and 2.

## Supplemental Information

Supplemental information for this article can be found online at http://dx.doi.org/10.7717/peerj.19911#supplemental-information.

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
