# Peer review of "Comparison of efficacy and costs between robotic-assisted and conventional thoracoscopic approaches for partial pulmonary resection: a systematic review and meta-analysis of propensity score-matched studies"

_PeerJ, doi:10.7717/peerj.19911_

## Round 0.1 · original submission · Major Revisions

**Language Note:** The review process has identified that the English language must be improved. PeerJ can provide language editing services - please contact us at [email protected] for pricing (be sure to provide your manuscript number and title). Alternatively, you should make your own arrangements to improve the language quality and provide details in your response letter. – PeerJ Staff

Reviewer 1 ·

Basic reporting

-

Experimental design

-

Validity of the findings

-

Additional comments

Dear Authors,
Thanks for your efforts in this NSCLC field.
I have some comments and suggestions for your manuscript.

1. The language is professional. However, there is a mistake in the title. Pneumonectomy or pulmonectomy? The meaning of those words is different. Pneumonectomy means the removal of the entire lung. Thus, you should modify it in the manuscript.

2. Literature references were provided.

3. The manuscript included a professional article structure, figures, and tables.

4. The first appearance of RAPP in the introduction has been specifically explained, and it is recommended to use the abbreviation directly in the later text. Please correct.

5. I think the article should make a clear scope of partial pulmonectomy, is it sublobectomy?

6. The biggest advantage of robotic-assisted surgery is lymph node dissection, but some lung resections do not require lymph node dissection. In addition, robot-assisted surgery takes too long because of the start-up time, and its advantages will only be realized when the operation is complex enough, such as complex segmentectomy and vascular sleeve anastomosis. These should be included in the discussion.

·

Basic reporting

-

Experimental design

-

Validity of the findings

-

Additional comments

I read with great interest the manuscript, which was about a meta-analysis performed with the aim of identifying differences in terms of cost and postoperative outcomes after partial lung resections performed either by VATS or RATS.

- This study included most of the new publications addressing this topic (up to March 2025), but these were mostly non-randomized studies.

- The aim of the study was clearly stated in the introduction section: to compare the outcomes and costs of VATS versus RATS.

- The inclusion and exclusion criteria were very detailed and precise in the selection of studies to be included in this manuscript.

- The total number of patients included was nearly 3,100, distributed almost equally between the two groups compared (VATS and RATS). The two groups also had the advantage of being comparable according to major demographic and clinical criteria.

Figures and tables:
- Please provide the corresponding references to each study in the respective tables.
- The expression of results in the tables should be refined. Express the variables and results of the studies presented in the tables as percentages (% of men, % conversions, etc.), and include the meaning of the values ​​given in the tables (e.g., means +/- eclipse, medians, etc.).

Some comments should be clarified:
- Why include Chen's 2022 study in your study when, according to the inclusion criteria and the risk of bias, this study had a critical level of bias?

- In the Results section, when including the names of the first authors of the studies, the authors should include the corresponding reference numbers. Furthermore, why didn't the authors provide only the results of the meta-analysis and the conclusions of the statistical tests performed on all eight studies included in this manuscript, and leave the discussion of the diversity or limitations of these studies to the Discussion section?

- References section: Is this reference placed at the end, referring to the first reference in the text?
"[1] Qiu T, Zhao Y, Xuan Y, Qin Y, Niu Z, Shen Y, Jiao W. Robotic sleeve lobectomy for centrally located non-small cell lung cancer: A propensity score-weighted comparison with thoracoscopic and open surgery. J Thorac Cardiovasc Surg. 2020 Sep;160(3):838-846.e2. doi: 0.1016/j.jtcvs.2019.10.158. Epub 2019 Nov 22. PMID: 31924355."

---

## Round 0.2 · accepted · Accept

We are pleased to inform you that your manuscript has been accepted for publication. We look forward to receiving your next manuscript.

With best regards,
Yoshi
Prof. Yoshinori Marunaka, M.D., Ph.D.